# Nonlinear RNNs as a Compute Shortcut for Time Series Foundation Models

**Levente Zólyomi** [1 2]  **David Stap** [2]  **Sebastian Böck** [2]  **Günter Klambauer** [1 2]  **Sepp Hochreiter** [1 2]

## Abstract

Time series foundation models have converged on Transformer and linear-RNN backbones that scale efficiently but do not provide an explicit nonlinear recurrent state-update mechanism, despite nonstationarity and long-range dependencies being central to time series data. Pure nonlinear recurrent models track state but scale poorly, suggesting an expressivity–efficiency trade-off. We show the trade-off is avoidable: adding *even a single* nonlinear sLSTM layer to three popular scalable backbones (Transformer, Gated DeltaNet, and mLSTM) consistently improves forecasting accuracy on GIFT-Eval across five parameter scales (1M–80M). The gain is largest at small scale, reaching roughly 4–5% for Transformer and Gated DeltaNet at 1M, and shrinks to under 1% for most 80M comparisons. Yet at matched wall-clock training budget, the 10M hybrid outperforms a same-sized baseline trained with $\geq 4\times$ more compute, and *a single* sLSTM placed early in the stack recovers most of the benefit gained from adding more state tracking layers. Nonlinear state tracking therefore functions as a compute shortcut: a cheap inductive bias that substitutes for capacity and training time.

## 1. Introduction

Time series foundation models (TSFMs) have rapidly matured into a distinct class of pretrained models, with large-scale benchmarks such as GIFT-Eval (Aksu et al., 2024) now enabling direct comparison across architectures. The field has converged on two architectural families: Transformers, as used in Chronos (Ansari et al., 2024), Moirai (Woo et al., 2024), and TimesFM (Das et al., 2024), and linear recurrent models derived from State Space Models (Gu & Dao, 2023) (FlowState (Graf et al., 2025)) and its alternatives such

as Gated DeltaNet (Yang et al., 2025) (Reverso (Fu et al., 2026), TempoPFN (Moroshan et al., 2025)). Both choices reflect architectural trends from large language modeling, where hardware efficiency and parallelizable pretraining have dominated design. Time series data differs from text in ways that these language-inherited choices may not address. Forecasting often requires handling nonstationarity, persistent hidden state, and long-range dependencies.

A growing literature identifies *state tracking*, the capacity to maintain and update a hidden state nonlinearly across time, as a core expressivity gap in modern sequence mixers: neither Transformers nor linear RNNs can track state in the sense of evaluating finite state machines or solving problems such as parity (Merrill et al., 2024; Grazzi et al., 2025). Recent work sharpens the theoretical argument: linear RNNs acquire expressive state tracking with modifications; negative eigenvalues (Grazzi et al., 2025) or Householder products (Siems et al., 2025), while Transformers handle state-tracking tasks empirically through learned shortcuts rather than expressible recurrences (Liu et al., 2023). Nonlinear recurrent architectures such as the LSTM (Hochreiter & Schmidhuber, 1997) retain this ability. The recently introduced xLSTM (Beck et al., 2024)—more specifically, its sLSTM cell—does as well. TiRex (Auer et al., 2025) shows this concretely: a 35M sLSTM-based TSFM matches or outperforms much larger Transformers on GIFT-Eval, establishing state tracking as a powerful inductive bias in forecasting. But scaling pure nonlinear recurrence sacrifices parallelism, creating an expressivity–efficiency trade-off.

We show this trade-off is mitigated by hybridizing: retain a scalable backbone and add nonlinear recurrence sparingly. We adopt the xLSTM $[n:m]$ notation (Beck et al., 2024) to describe hybrids of $n$ non-state-tracking layers to $m$ sLSTM layers. Hybridization has recent precedent in language modeling, where several large-scale systems interleave Transformer with linear-RNN or state-space layers (Lieber et al., 2024; Glorioso et al., 2024; Ren et al., 2024; NVIDIA, 2025; Qwen Team, 2025); our move is distinctive in that the sparsely added layer supplies the nonlinear state-tracking capability that both Transformer and linear RNN backbones lack. We evaluate sLSTM hybrids across three scalable backbones (Transformer, Gated DeltaNet, mLSTM (Beck et al., 2024)), five parameter scales (1M–80M), and two training regimes (parameter-matched and

[1] Johannes Kepler University, Linz, Austria [2]NXAI GmbH, Linz, Austria. Correspondence to: Levente Zólyomi <levente.zolyomi@nx-ai.com>.

*Proceedings of the $2^{nd}$ ICML Workshop on Foundation Models for Structured Data*, Seoul, South Korea. 2026. Copyright 2026 by the author(s).

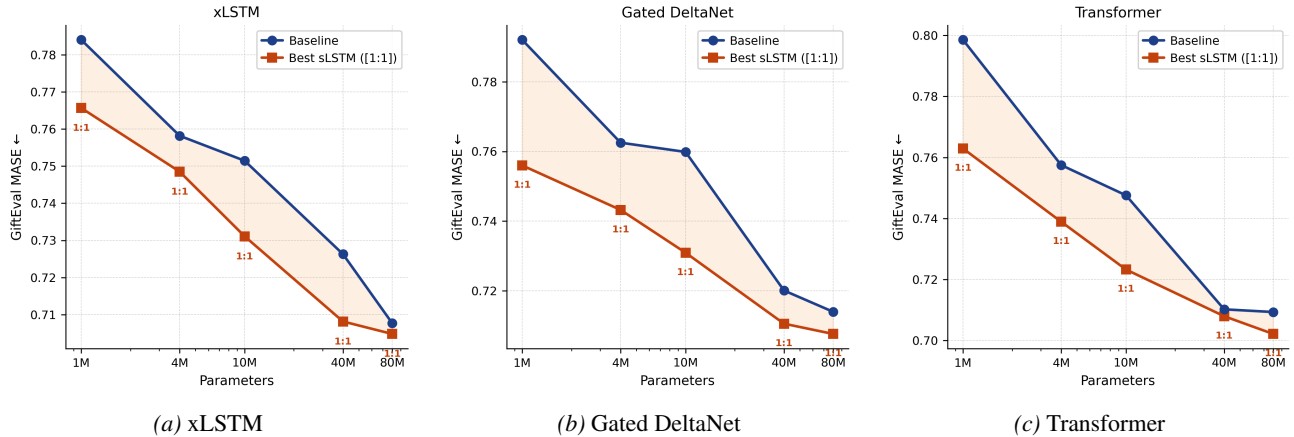

*(a)* xLSTM          *(b)* Gated DeltaNet          *(c)* Transformer

*Figure 1.* **Hybrids beat pure baselines at every scale, but the gap narrows with scale.** GIFT-Eval MASE (lower is better) against parameter count for three backbones, each in pure baseline and hybrid $[1:1]$ form. The relative gap shrinks from 5% at 1M to under 1% at 80M, indicating that higher capacity can absorb the inductive bias introduced via state tracking.

wall-clock-matched) on GIFT-Eval, finding that the hybrid's gain over each baseline is consistent but scale-dependent, that it persists under matched compute, and that a *single* sLSTM layer placed early in the stack is sufficient. We call this a compute shortcut: an architectural bias that achieves, at fixed parameter count or wall-clock budget, performance that a pure scalable backbone would require substantially more capacity or training time to reach.

**Contributions.** We show that:

- **Sparse nonlinear recurrence consistently improves scalable backbones.** Adding sLSTM layers improves Transformer, Gated DeltaNet, and mLSTM baselines across five parameter scales on GIFT-Eval. The gain is around 5% at 1M and diminishes to under 1% at 80M.

- **The diminishing gap reflects an inductive bias being absorbed, not a failure to scale.** At matched parameter count, larger backbones close the gap; at matched wall-clock training, a 10M hybrid outperforms a 10M pure baseline trained with $\geq 4\times$ more compute. The inductive bias provides a more efficient route to the same accuracy than increasing capacity or training time.

- **A single sLSTM layer placed early suffices.** One sLSTM near the input of a 6-layer stack recovers most of the benefit of the full $[1:1]$ hybrids, therefore replacing a single layer alone can suffice to add state-tracking capabilities to a backbone.

## 2. Experimental Setup

We evaluate on GIFT-Eval (Aksu et al., 2024), a benchmark of 23 datasets across 7 domains with multiple frequencies and forecasting horizons, and report the geometric mean MASE and CRPS aggregated across datasets. We follow

the GIFT-Eval protocol for forecasting horizons, splits, and evaluation. We exclude GIFT-Eval training datasets from pretraining, so all reported evaluation is zero-shot.

We consider three scalable backbones as the non-state-tracking family: Transformer, for its dominance among current TSFMs (Ansari et al., 2024; 2025; Das et al., 2024; Woo et al., 2024); Gated DeltaNet (Yang et al., 2025), a representative linear RNN used in Fu et al. (2026) and its Gated DeltaProduct extension (Siems et al., 2025) in Moroshan et al. (2025); and mLSTM, a linear RNN variant which shares the xLSTM lineage (Beck et al., 2024) with the sLSTM state-tracking layer. For Gated DeltaNet we allow negative eigenvalues in the state-transition matrix following Grazzi et al. (2025), with an ablation showing its effect in Appendix A. Our state-tracking layer is sLSTM (Beck et al., 2024), a variant of the LSTM (Hochreiter & Schmidhuber, 1997) with exponential gating.

Following the notation of Beck et al. (2024), we write $[n:m]$ for an architecture with $n$ non-state-tracking layers per $m$ sLSTM layers; baselines are pure Transformer, pure Gated DeltaNet, pure xLSTM$[1:0]$ [1] and our main hybrid is $[1:1]$, with placement and density variations further explored in Section 3.3.

We sweep five parameter scales from 1M to 80M by varying width and depth; Table 1 gives the configuration at each scale. The smallest model replaces the residual-MLP input/output projection used at larger scales with a linear projection to keep the 1M parameter budget tight. All backbones at a given scale share the same width, depth, and MLP expansion factor. For xLSTM and Gated DeltaNet baselines we keep the short convolution used in their original formu-

---

[1] We follow the xLSTM paper in using xLSTM$[1:0]$ to denote a pure mLSTM architecture.

lations. For hybrids we disable this convolution, because an ablation in Appendix B shows that it does not improve hybrid performance and often degrades it while improving pure baselines.

Pretraining data for the models combines subsets of the Chronos (Ansari et al., 2024) and GIFT-Eval-Pretrain (Aksu et al., 2024) corpora, augmented with synthetic sequences from KernelSynth (Ansari et al., 2024) and the augmentation pipeline introduced in TiRex (Auer et al., 2025). Sequences are tokenized into non-overlapping patches of 32 timesteps and embedded via a residual MLP block, as is standard in TSFMs (Ansari et al., 2024; Das et al., 2024; Auer et al., 2025). We optimize the models on the pinball loss (Auer et al., 2025; Ansari et al., 2025) with AdamW (Loshchilov & Hutter, 2019) at learning rate $1.2 \times 10^{-3}$, cosine schedule with 3% warmup, effective batch size 256, context length 8192, for 500,000 optimization steps on a single $4 \times$A100 node.

| Scale | Width | Depth | In/Out proj. | MLP exp. |
|---|---|---|---|---|
| 1M | 128 | 6 | linear | 2.0 |
| 4M | 256 | 6 | MLP, 1024 | 2.0 |
| 10M | 384 | 6 | MLP, 2048 | 2.0 |
| 40M | 512 | 12 | MLP, 2048 | 2.75 |
| 80M | 768 | 12 | MLP, 2048 | 2.75 |

*Table 1.* **Architecture hyperparameters for the five parameter scales used throughout.** Width (hidden dimension), depth (number of layers), input/output projection ("linear" at 1M, residual-MLP with the listed hidden dimension otherwise), and MLP expansion factor. All backbones at a given scale share these settings.

## 3. Results

### 3.1. Hybrids beat pure baselines, shrinking at scale

Across all three backbones and all five scales, the $[1:1]$ hybrid outperforms its pure $[1:0]$ baseline on GIFT-Eval (Figure 1). The relative gap decreases monotonically with parameter count, narrowing from 5% at 1M to under 1% at 80M. The effect is not backbone-specific: it persists across families with different inductive biases, including the strongest baseline (Transformer). This is consistent with the hypothesis that sLSTM supplies a nonlinear state-tracking bias that all three backbones otherwise lack, rather than patching a weakness of any one of them. Decomposing by forecast horizon (Table 2; full breakdown in Appendix C, Table 6), gains are largest at long horizons, substantial at medium, and vanish at short horizons - indicating that state tracking's advantage emerges mostly in long horizon forecasts.

This scale-dependent narrowing has only a directional precedent in language modeling, where hybrid Transformer/linear-RNN systems are by now standard but

typically reported at a single scale with sparse-attention ratios: roughly 1:7 in Jamba (Lieber et al., 2024), 4 of 52 in Nemotron-H (NVIDIA, 2025), and 7% attention in the Mamba-2-Hybrid study (Waleffe et al., 2024); Hauzenberger et al. (2026) provide xLSTM-family precedent by distilling hybrid xLSTM students that recover most of their quadratic-attention teachers' performance. The closest match for our gap-vs-scale finding is Olmo Hybrid (Merrill et al., 2026), which reports hybrid–pure gaps closing during training, directionally mirroring our monotonic narrowing across parameter count.

These patterns are consistent with Liu et al. (2023), who show that Transformers at sufficient scale learn algorithmic shortcuts to state-tracking problems they cannot express in closed form: the hybrid's advantage is largest where the baseline has neither the capacity to learn such shortcuts nor the inductive bias. In our experiments, at 80M the baseline and hybrid curves nearly converge, setting up the next question: does the hybrid retain an advantage when compute, rather than parameter count, is held fixed?

| Size | Model | Aggregate GIFTEval | | Forecast Horizon (MASE) | | |
|---|---|---|---|---|---|---|
| | | MASE↓ | CRPS↓ | long↓ | medium↓ | short↓ |
| 1M | xLSTM[1:0] | 0.784 | 0.542 | 0.877 | 0.834 | 0.734 |
| | xLSTM[1:1] | 0.766 | 0.525 | 0.815 | 0.794 | 0.737 |
| 4M | xLSTM[1:0] | 0.762 | 0.521 | 0.843 | 0.799 | 0.720 |
| | xLSTM[1:1] | 0.749 | 0.517 | 0.800 | 0.769 | 0.722 |
| 10M | xLSTM[1:0] | 0.751 | 0.517 | 0.821 | 0.781 | 0.716 |
| | xLSTM[1:1] | 0.731 | 0.503 | 0.768 | 0.740 | 0.714 |
| 40M | xLSTM[1:0] | 0.718 | 0.490 | 0.765 | 0.734 | 0.695 |
| | xLSTM[1:1] | 0.708 | 0.484 | 0.744 | 0.717 | 0.692 |
| 80M | xLSTM[1:0] | 0.708 | 0.487 | 0.750 | 0.716 | 0.689 |
| | xLSTM[1:1] | 0.705 | 0.485 | 0.741 | 0.710 | 0.690 |

*Table 2.* **Across short, medium, and long horizons, hybrids' advantage grows with horizon length.** xLSTM$[1:0]$ vs xLSTM$[1:1]$ across all five parameter scales on GIFT-Eval; lower is better.

### 3.2. Matched-compute advantage

We next test whether the advantage persists when compute, rather than parameter count, is held fixed. We train Transformer and xLSTM backbones in pure $[1:0]$ and hybrid $[2:1]$ form at 10M parameters under three wall-clock budgets (2.5, 5 and 10 hours), measured in node-hours on a $4 \times$A100 node (excluding data loading and evaluation).

At every budget, the hybrid dominates its pure counterpart (Figure 2). At the lowest budget, the $[2:1]$ hybrid already reaches MASE levels that the pure $[1:0]$ baseline does not reach even at the highest budget, corresponding to a factor of $\geq 4\times$ in training compute.

This holds despite the pure Transformer baseline receiving roughly 50% more optimization steps at matched wall-clock due to faster attention kernels at our scale, an effect that

works against, not for, the hybrid's advantage. The baseline takes more optimization steps per node-hour and still loses: this implies that the inductive bias of nonlinear state tracking cannot be compensated for with a larger training budget.

Compute-matched training comparisons across TSFM architectures remain largely unexplored. The closest precedent in language modeling is Griffin (De et al., 2024), which reports matching Llama-2 on 6× fewer training tokens, a per-token rather than per-wall-clock comparison. Most hybrid language-model work emphasizes inference-time efficiency (KV-cache savings in Jamba; faster decoding in Nemotron-H) rather than the pretraining wall-clock budget studied here.

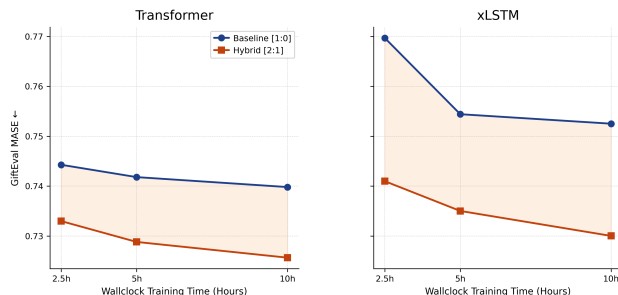

*Figure 2.* **The hybrid's advantage persists under matched compute.** GIFT-Eval MASE at 10M parameters as a function of wall-clock training budget (node-hours on 4×A100) for Transformer and xLSTM backbones in pure $[1:0]$ and hybrid $[2:1]$ form. At matched compute the hybrid dominates; at the lowest budget it already matches the accuracy the pure baseline reaches only at the highest budget, a factor of $\geq 4\times$ in training compute, even though attention kernels afford the baseline roughly 50% more optimization steps per node-hour.

### 3.3. One layer suffices

We next ask how many sLSTM layers are needed, and where they should be placed. Fixing the stack depth at six layers at 10M parameters in the xLSTM family, we vary which positions use sLSTM (Table 3).

A single sLSTM at position 1 (config 1) recovers roughly 75% of the gain obtained by the densest configuration (config 8, four sLSTMs in six layers), while adding more sLSTMs yields smaller additional improvements. Placement also matters: the same single sLSTM moved to position 2 or 5 (configs 2, 3) captures a smaller fraction of the gain than at position 1, and configurations that interleave sLSTMs across early and late layers (e.g. config 7) outperform configurations that cluster them (config 4). One sLSTM early in the stack can therefore be sufficient in practice, and the hybrid architectures do not need to be dense in recurrent layers to secure its advantage.

Similar layer-ratio ablations have only been conducted for hybrid language models and those have converged on sparse-

attention recommendations: Jamba (Lieber et al., 2024) finds no gap between 1:3 and 1:7 ratios, and Waleffe et al. (2024) settle on about 7% attention in an 8B hybrid. These works do not ablate *which* layers carry the sparse specialty component. Our placement result is complementary: where the sparse layer sits matters as much as how many there are, with early placement dominating. Whether the input-side preference generalizes beyond TSFMs, i.e. whether hybrid language models benefit from front-loaded attention, is an open question.

| Config | \multicolumn{6}{c}{Layer position} | # sLSTM | MASE ↓ (mean ± std) |
| | 0 | 1 | 2 | 3 | 4 | 5 | | |
|---|---|---|---|---|---|---|---|---|
| 0 | | | | | | | 0 | 0.7606 |
| 1 | | ● | | | | | 1 | 0.7364 ± 0.0026 |
| 2 | | | ● | | | | 1 | 0.7415 ± 0.0012 |
| 3 | | | | | | ● | 1 | 0.7404 ± 0.0041 |
| 4 | | ● | ● | | | | 2 | 0.7414 ± 0.0025 |
| 5 | | | ● | | | ● | 2 | 0.7359 ± 0.0018 |
| 6 | | | | | ● | ● | 2 | 0.7359 ± 0.0032 |
| 7 | | ● | | ● | | ● | 3 | 0.7331 ± 0.0026 |
| 8 | ● | ● | | ● | ● | ● | 4 | 0.7312 ± 0.0041 |

*Table 3.* **A single sLSTM near the input recovers most of the** $[1:1]$ **benefit.** Placement ablation at 10M xLSTM (6 blocks). ● marks positions where sLSTM replaces mLSTM. Config 0 is the pure xLSTM$[1:0]$ baseline; configs 1–3 use a single sLSTM at different depths; configs 4–8 add progressively more sLSTMs.

## 4. Conclusion

Sparse nonlinear recurrent layers, which provide a natural state-tracking mechanism, act as a compute shortcut for time series foundation models: adding even a single sLSTM layer to Transformer, Gated DeltaNet, or mLSTM backbones consistently improves forecasting performance of TSFMs on GIFT-Eval across five parameter scales, and the advantage persists when compute rather than parameter count is held fixed. The shrinking gap at scale is consistent with an inductive bias being absorbed by capacity, rather than with a scaling ceiling: at matched wall-clock training, the 10M hybrid still reaches accuracies that pure baselines require $\geq 4\times$ more compute to match.

Framing nonlinear recurrence as a compute shortcut, rather than a competing backbone, reframes TSFM design from a question of *which architecture wins* to one of *which inductive biases to inject into a scalable substrate*—a more economical use of compute than scaling pure recurrent or pure attention stacks. Extending this approach to other forecasting settings where state tracking is salient, such as longer contexts or streaming inference, is an interesting direction for future work.

## Acknowledgements

We acknowledge the EuroHPC Joint Undertaking for awarding us access to Leonardo at CINECA, Italy.

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

## A. Negative eigenvalues in Gated DeltaNet

In our study we use Gated DeltaNet with negative eigenvalues enabled in the state transition matrix as described in Grazzi et al. (2025). Since the core idea behind these negative eigenvalues is improved state tracking, in this section we ablate their effect on different models in 4M and 10M scales. We present results in Table 4. Across both scales we consistently observe that negative eigenvalues in the state transition matrix improve performance of the models. Notably, for baseline models the improvement is $\geq 5\%$, while for hybrids the improvements are more modest, often within $1\%$, implying that the improved state tracking enabled by the negative eigenvalues is partly subsumed by the sLSTM layers.

| Metric | Condition | GDN-4M | GDN[1:1]-4M | GDN[1:1]-10M | GDN-10M |
|--------|-----------|--------|-------------|--------------|---------|
| MASE | No negative eigenvalues | 0.8260 | 0.7466 | 0.7334 | 0.817 |
| | With negative eigenvalues | **0.7822** | **0.7432** | **0.7309** | **0.764** |
| CRPS | No negative eigenvalues | 0.5653 | 0.5165 | **0.5022** | 0.561 |
| | With negative eigenvalues | **0.5323** | **0.5153** | 0.5030 | **0.517** |

*Table 4.* MASE and CRPS for GDN variants with and without negative eigenvalues. Lower is better; best per column per metric in bold.

## B. Effect of convolution in linear RNNs

Both xLSTM and Gated DeltaNet utilize short convolutions before the main sequence mixing operation. While their effect is established in language modeling, for time series their behavior is not well understood, therefore we present a short ablation to determine their effect in our study. Table 5 presents these results. The short convolutions consistently improve the performance on the baseline models, however they present an adverse effect in hybrids, potentially interfering with the dynamics of the sLSTM layers, therefore we turn convolutions off in our hybrids.

| Metric | Condition | xLSTM[1:0]-4M | xLSTM[2:1]-4M | GDN-4M | GDN[2:1]-4M | xLSTM[1:0]-10M | GDN-10M |
|--------|-----------|---------------|---------------|--------|-------------|----------------|---------|
| MASE | No conv | 0.7753 | **0.7388** | 0.7987 | **0.7418** | 0.7606 | 0.7815 |
| | With conv | **0.7620** | 0.7500 | **0.7657** | 0.7452 | **0.7515** | **0.7636** |
| CRPS | No conv | 0.5330 | **0.5092** | 0.5432 | **0.5099** | 0.5219 | 0.5307 |
| | With conv | **0.5210** | 0.5156 | **0.5214** | **0.5099** | **0.5167** | **0.5176** |

*Table 5.* MASE and CRPS for xLSTM and GDN variants with and without convolution. Lower is better; best per column per metric in bold. For baseline models convolutions consistently improve performance across architectures and scales. For hybrid models, convolutions do not provide a consistent benefit and often degrade MASE, so we report hybrids with convolutions disabled.

## C. Extended Results

| Size | Model | Generic | | Forecast Horizon (MASE) | | |
|---|---|---|---|---|---|---|
| | | MASE ↓ | CRPS ↓ | long ↓ | medium ↓ | short ↓ |
| | Transformer | 0.799 | 0.537 | 0.910 | 0.869 | 0.736 |
| | Transformer[2:1] | 0.756 | 0.517 | 0.809 | 0.773 | 0.730 |
| | Transformer[1:1] | 0.763 | 0.525 | 0.820 | 0.792 | 0.731 |
| | GDN | 0.792 | 0.541 | 0.943 | 0.832 | 0.727 |
| 1M | GDN[2:1] | 0.762 | 0.522 | 0.817 | 0.789 | 0.733 |
| | GDN[1:1] | 0.758 | 0.521 | 0.812 | 0.779 | 0.730 |
| | xLSTM[1:0] | 0.784 | 0.542 | 0.877 | 0.834 | 0.734 |
| | xLSTM[2:1] | 0.768 | 0.527 | 0.828 | 0.802 | 0.735 |
| | xLSTM[1:1] | 0.766 | 0.525 | 0.815 | 0.794 | 0.737 |
| | Transformer | 0.766 | 0.522 | 0.865 | 0.807 | 0.717 |
| | Transformer[2:1] | 0.751 | 0.516 | 0.816 | 0.776 | 0.718 |
| | Transformer[1:1] | 0.750 | 0.514 | 0.819 | 0.779 | 0.716 |
| | GDN | 0.766 | 0.521 | 0.861 | 0.806 | 0.718 |
| 4M | GDN[2:1] | 0.742 | 0.510 | 0.795 | 0.765 | 0.714 |
| | GDN[1:1] | 0.745 | 0.516 | 0.802 | 0.764 | 0.717 |
| | xLSTM[1:0] | 0.762 | 0.521 | 0.843 | 0.799 | 0.720 |
| | xLSTM[2:1] | 0.739 | 0.509 | 0.786 | 0.761 | 0.713 |
| | xLSTM[1:1] | 0.749 | 0.517 | 0.800 | 0.769 | 0.722 |
| | Transformer | 0.748 | 0.509 | 0.819 | 0.781 | 0.710 |
| | Transformer[2:1] | 0.731 | 0.501 | 0.774 | 0.747 | 0.710 |
| | Transformer[1:1] | 0.723 | 0.495 | 0.763 | 0.734 | 0.705 |
| | GDN | 0.764 | 0.517 | 0.886 | 0.815 | 0.704 |
| 10M | GDN[2:1] | 0.727 | 0.502 | 0.771 | 0.743 | 0.705 |
| | GDN[1:1] | 0.732 | 0.503 | 0.780 | 0.750 | 0.708 |
| | xLSTM[1:0] | 0.751 | 0.517 | 0.821 | 0.781 | 0.716 |
| | xLSTM[2:1] | 0.733 | 0.508 | 0.774 | 0.744 | 0.713 |
| | xLSTM[1:1] | 0.731 | 0.503 | 0.768 | 0.740 | 0.714 |
| | Transformer | 0.710 | 0.484 | 0.752 | 0.725 | 0.689 |
| | Transformer[3:1] | 0.709 | 0.487 | 0.754 | 0.719 | 0.689 |
| | Transformer[1:1] | 0.708 | 0.486 | 0.745 | 0.719 | 0.690 |
| | GDN | 0.720 | 0.496 | 0.783 | 0.744 | 0.689 |
| 40M | GDN[3:1] | 0.716 | 0.492 | 0.754 | 0.726 | 0.698 |
| | GDN[1:1] | 0.711 | 0.487 | 0.749 | 0.716 | 0.695 |
| | xLSTM[1:0] | 0.718 | 0.490 | 0.765 | 0.734 | 0.695 |
| | xLSTM[3:1] | 0.716 | 0.493 | 0.755 | 0.723 | 0.699 |
| | xLSTM[1:1] | 0.708 | 0.484 | 0.744 | 0.717 | 0.692 |
| | Transformer | 0.709 | 0.486 | 0.760 | 0.723 | 0.686 |
| | Transformer[3:1] | 0.705 | 0.482 | 0.747 | 0.716 | 0.686 |
| | Transformer[1:1] | 0.702 | 0.482 | 0.736 | 0.712 | 0.686 |
| | GDN | 0.714 | 0.489 | 0.771 | 0.725 | 0.689 |
| 80M | GDN[3:1] | 0.706 | 0.482 | 0.748 | 0.715 | 0.688 |
| | GDN[1:1] | 0.708 | 0.484 | 0.748 | 0.712 | 0.691 |
| | xLSTM[1:0] | 0.708 | 0.487 | 0.750 | 0.716 | 0.689 |
| | xLSTM[3:1] | 0.715 | 0.490 | 0.756 | 0.730 | 0.694 |
| | xLSTM[1:1] | 0.705 | 0.485 | 0.741 | 0.710 | 0.690 |

*Table 6.* Extended results on all backbones, across all scales and multiple configurations.

