# OpenReview forum: "Nonlinear RNNs as a Compute Shortcut for Time Series Foundation Models"
_ICML.cc/2026/Workshop/FMSD — FMSD @ ICML 2026 Poster_

### Official Review · Reviewer_d21G · 2026-05-14
**Interesting findings regarding the addition of  sLSTM layers to time series backbones**

**Rating:** 7
**Confidence:** 5

**Review:**

Summary
The paper addresses the problem of adding sLSTM layers to time series back bones to introduce inductive bias that lead to better accuracy with less parameters or less compute. Multiple backbones are evaluated, and the introduction of slstm layers improve the performance for all architectures and all parameters sizes

Strengths
The paper is clear and easy to read, claims are backed by experiemental evidece, multiple archiectures and back bones are evaluated motivating the benefits of the slstm layer

Areas for Improvement

The results would be me more convincing if applied to state of the art TSFM models

Detailed comments:

Some other limitations include

1) The comparison may not fully isolate the effect of sLSTM
The hybrids differ from the baselines in more than just “state tracking”. For example, the paper disables short convolutions in hybrids because they hurt hybrid performance, while keeping them for some pure baselines

2) The approach is evaluated on a single dataset

Score Justification

I recommend acceptance because the paper presents a clear, well-motivated, and practically relevant contribution. The idea of introducing sparse sLSTM layers as a lightweight inductive bias for TSFMs is interesting, and the paper supports its claims with experiments across multiple architectures, parameter scales, and compute settings. The paper is also well written and easy to follow.

In particular, the matched-compute experiments and layer placement ablations strengthen the contribution beyond a simple architecture comparison. The consistency of the improvements across different backbones suggests that the proposed approach is broadly applicable.

While there are some limitations, including evaluation on a single benchmark and comparisons that do not fully isolate the effect of the sLSTM layers from other architectural modifications, these issues do not outweigh the overall quality and relevance of the contribution. The paper provides useful empirical insights for TSFM design and would likely be of interest to the community.

---

### Official Review · Reviewer_GQMq · 2026-05-18
**Nonlinear RNNs as a Compute Shortcut for Time Series Foundation Models**

**Rating:** 6
**Confidence:** 3

**Review:**

### **Summary:**
- This paper investigates sparse nonlinear recurrence can improve scalable TSFM backbones across architectures and parameter scales. Empirical findings support that adding a small number of sLSTM layers improves GIFT-Eval forecasting performance, especially at smaller scales and under matched compute.

### **Strengths:**
- The paper addresses an important TSFM design question: whether sparse nonlinear recurrence can improve scalable backbones for forecasting.
- Experiments across three backbones and five parameter scales show consistent improvements from adding sLSTM layers.
- The matched-compute and layer-placement analyses provide useful practical insights for compute-efficient TSFM design.

### **Weakness:**
- The compute-shortcut claim is mainly supported at limited matched-compute settings, without broader inference-cost analysis.
- The gains shrink substantially at larger scales where most TSFMs parameter size are larger than 80M, making the practical benefit of adding sLSTM layers less clear.
- The methodological novelty is moderate, as the approach mainly hybridizes existing backbones with existing sLSTM layers.

### **Suggestions:**
- Comparing with existing TSFMs such as Chronos, Moirai, TimesFM would be helpful for a comparison against hybrid model

---

### Official Review · Reviewer_9EHB · 2026-05-22

**Rating:** 6
**Confidence:** 5

**Review:**

This paper introduces the idea that adding a single sparse nonlinear sLSTM layer early in scalable models (Transformers, linear RNNs) acts as a "compute shortcut," improving time series forecasting. The approach is practical, requires minimal modification, and shows consistent gains, especially at smaller scales, with benefits diminishing at larger sizes. The extensive experiments demonstrate that this inductive bias effectively enhances performance under both parameter and compute constraints. However, the benefits reduce as models grow, and a deeper theoretical understanding would strengthen the work.